# Quality and Health Risk Assessment of Groundwater for Drinking and Irrigation Purpose in Semi-Arid Region of India Using Entropy Water Quality and Statistical Techniques

**Balamurugan Panneerselvam** [1,*] **, Nagavinothini Ravichandran** [2]**, Shunmuga Priya Kaliyappan** [3]**,**
**Shankar Karuppannan** [4] **and Butsawan Bidorn** [1,5,*]

1   Center of Excellent of Interdisciplinary Research for Sustainable Development, Chulalongkorn University, Bangkok 10330, Thailand
2   Department of Structures for Engineering and Architecture, University of Naples Federico II, 80125 Naples, Italy
3   Department of Civil Engineering, Nehru Institute of Technology, Coimbatore 641105, India
4   Department of Applied Geology, School of Applied Natural Sciences, Adama Science and Technology University, Adama 1888, Ethiopia
5   Department of Water Resources Engineering, Chulalongkorn University, Bangkok 10330, Thailand
*   Correspondence: balamurugan.phd10@gmail.com (B.P.); butsawan.p@chula.ac.th (B.B.)

**Abstract:** The continuous intake of contaminated drinking water causes serious issues for human health. In order to estimate the suitability of groundwater for drinking and irrigation, and also conduct human risk assessments of various groups of people, a total of 43 sample locations in the semi-arid southern part of India were selected based on population density, and we collected and analyzed groundwater from the locations for major anions and cations. The present study's novelty is integrating hydrochemical analysis with the entropy water quality index (EWQI), nitrate pollution index (NPI) and human health risk assessment. The results of the EWQI revealed that 44.19% of the sample locations need to be treated before consumption. About 37.20% of the study region has a high concentration of nitrate in the groundwater. NPI revealed that 41.86% of the samples had moderate or significant pollution levels. The non-carcinogenic risk evaluation showed that 6–12-year-old children are at a higher risk than teenagers, adults and elderly people in the study area. The natural sources of nitrate and other contamination of groundwater are rock–water interaction, weathering of rock, dissolution of carbonate minerals and evaporation processes, and the anthropogenic sources are the decomposition of organic substances in dumping yards, uncovered septic tanks and human and animal waste. The results suggest taking mitigation measures to reduce the contamination and improve the sustainable planning of groundwater management.

**Keywords:** groundwater; human health; nitrate pollution index; nitrate contamination; noncarcinogenic risk



## 1. Introduction

Groundwater is the major source of water for drinking and irrigation purposes all around the world [1–3]. Developed and developing countries such as the United States of America, Poland, Italy, China, India and Pakistan majorly depend on groundwater for drinking and irrigation. Particularly in India, high percentages of people living in rural and urban areas rely on groundwater for their daily needs [4–7]. Accordingly, there is a significant need to investigate groundwater quality before consumption. Water contamination due to increased industrialization and urbanization is a serious threat to human health and agriculture [8–10]. Groundwater gets contaminated in two ways: from natural sources such as a floods, excess amounts of mineral present in the aquifers, landslides and rainfall leachates, and from anthropogenic sources such as the disposal of municipal waste, usage of synthetic fertilizers in agriculture fields and industrial waste disposal [11–15].

Recently, researchers and scientists [16–24] have become more involved in assessing groundwater quality for domestic and agricultural purposes. Previous studies indicated that anthropogenic activities highly affect groundwater quality in arid and semi-arid regions worldwide [25–28]. Recent studies that were carried out in semi-arid regions of India showed that improper waste management, municipal waste disposal, uncovered septic tanks, leakages in water lines and usage of chemical fertilizers and pesticides are the major sources of groundwater contamination [29,30].

Among all the contaminants, nitrate and fluoride are highly percolated into aquifer systems as they originate from various geogenic and non-geological activities [31–33]. Environmental conditions such as low precipitation, high evaporation and infiltration of leachates from waste disposal yards enhance the water salinity and also increase the toxicity of certain chemicals (such as nitrate) in groundwater. Continuously consuming contaminated groundwater causes various diseases, presenting carcinogenic and non-carcinogenic risks to the human body [5,34,35]. An increase in groundwater contamination leads to serious health problems in the human body and for other living organisms. The United States Environmental Protection Agency [36] developed regulations and a methodology to assess the human health risk due to various groundwater contaminants for two major exposure pathways: oral and dermal contact. Panneerselvam et al. [4] assessed nitrate contamination and its impact on human health in the semi-arid southern part of India, and the results revealed that the decomposition of organic substances and leachates from dumping yards is the primary source of contaminants in groundwater. Ramalingam et al. [37] evaluated the impact of excess nitrate concentrations in groundwater and described how the use of synthetic fertilizers, improper maintenance of underground pipelines and uncovered septic tanks are the main sources of excess nitrate in groundwater.

The EWQI is an effective method to represent the chemical composition of water in a single value. It is an advanced and accurate method to evaluate groundwater quality using the entropy value of each water quality parameter. The weightage assigned to each parameter depends on the importance of the parameter in the chemical composition of water. The EWQI classifies water quality into five classes: excellent, good, medium, poor and very poor for drinking and domestic usage. Egbueri [38] conducted a detailed investigation of the soft computing model, to incorporate entropy theory with an artificial neural network. The results revealed that the EWQI and integrated EWQI have good accuracy for predicting each sample location's water quality index value. Kumar and Augustine [39] assessed groundwater quality modeling using the EWQI and spatial techniques, and the study identified that the chemicals NaCl and Ca-Mg-Cl most affected water in the investigation zone.

Another factor directly and indirectly affecting groundwater quality is seasonal variability, including rainfall intensity, temperature variation and humidity of atmospheres [40,41]. These are the primary factors that affect the chemical composition of groundwater during the process of evaporation during summer (dry season) and infiltration of rainfall water during rainy days (post-monsoon season) [42]. In the present research, the study area is a semi-arid region with high seasonal variability due to the temperature and evaporation.

Based on previous studies, the specific objectives of the present research were (1) to assess the groundwater characteristics based on the World Health Organization criteria [43], (2) to evaluate the suitability of groundwater for domestic usage, (3) to calculate the nitrate pollution index and conduct a human health risk assessment and (4) to statistically determine the contamination sources in the study region. The novelty of the present study was its integrated approach to assessing the contaminants in groundwater. This was also the first study carried out in the research region to assess the groundwater quality for drinking and irrigation purposes. We proceeded by measuring the EWQI of groundwater samples, conducting a nitrate contamination assessment aided by the NPI and completing a human health risk assessment among various groups of people. The result of the present study will help enhance sustainable practices in groundwater management to avoid contamination in the study region.

## 2. Materials and Methods

### 2.1. Study Area Description

Salem is one of the fast-developing districts in Tamil Nadu, South India. The study area covered an area of 611.72 sq.km and is located in the northeast part of the Salem district. It is bounded by the Perambalur district on the west, Gangavalli Taluk on the south and Pethanayakanpalayam Taluk on the east (Figure 1). The geological coordinates of the study area are $11°26'30''$ N–$11°41'30''$ N latitude and $78°26'16''$ E–$78°49'50''$ E longitude. The study region has a high number of sago factories and small-scale industries. A high percentage of people in the study region depend on groundwater for drinking and irrigation purposes. The study region's climate is hotter from February to June (37–44 °C) and colder from November to January (18–22 °C). Under the influence of southwesterly and northeasterly monsoons, the study region receives precipitation. However, on the whole, the study area is categorized as a dry and semi-arid region. The major businesses in the study area are industrial and agricultural work. The study region's major crops frequently cultivated are sugarcane, cotton, groundnut, gingelly, oilseeds, millet and rice.

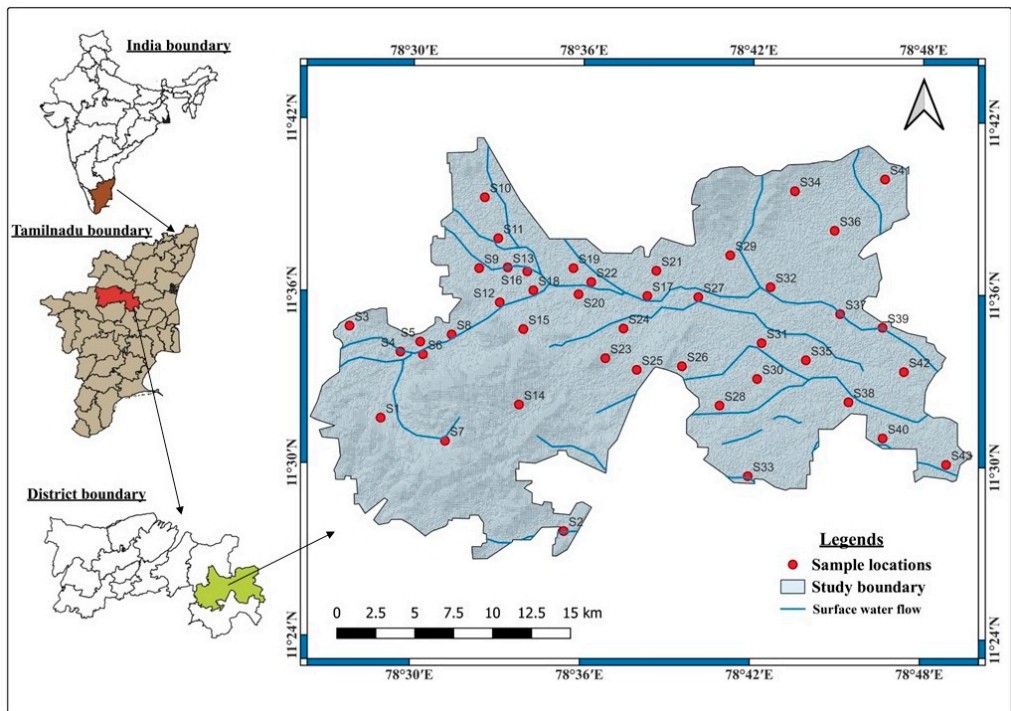

**Figure 1.** Groundwater sample locations in study region.

### 2.2. Geology and Hydrology

The study area is mainly comprised of recent alluvium deposits with a Crystalline Archaean formation along the inter-mountain valley. A high percentage of the study region is covered by biotite gneiss, charnockite, porphyroclasts bearing mylonites and gneisses. These rocks belong to the gneissic and charnockite groups. The western part of the study region is covered by magnetite-quartzite, amphibolite and met sediments. During the preliminary survey, the research team found that gneisses are highly weathered and display fractures and joints. Moreover, soil investigation revealed that landforms, highly weathered calcareous clay soil and moderately shallow clay soil can be found in the entire research region.

### 2.3. Methods

#### 2.3.1. Sample Collection and Analysis

Groundwater samples were collected from densely populated and agriculture field study areas. A total of 43 samples were collected during the post-monsoon season before

the COVID-19 pandemic period (2018) from bore wells/hand pumps based on the available sources of groundwater (Figure 1). A 1 L quantity of sample was collected in high-density polythene bottles. The sample container was prewashed with $HNO_3$ acid and distilled water 2 to 3 times before use [44,45]. The standard procedure was followed during the sample collection, such as bore wells being pumped for 5–10 min prior to the sample collection to avoid the influence of accumulated water in the pipeline. The collected samples were transferred to the laboratory and kept at 4 °C until analyzed [46,47]. The physical characteristic of groundwater, such as pH and EC, were calculated during the sample collection using an Elico pH meter and conductivity meter. The chemical characteristics of samples such as calcium, magnesium, chloride and bicarbonate were calculated using volumetric titration methods. Meanwhile, the sodium and potassium in each water sample were estimated using a flame photometer. Sulfate in the samples was recorded using a spectrophotometer by following the American Public Health Association recommendation [48]. The recorded values were checked with the ionic balance error (IBE) method to determine the accuracy of the analytical results using Equation (1):

$$\text{IBE}(\%) = \frac{\sum cations - \sum anions}{\sum cations + \sum anions} \times 100 \tag{1}$$

Note: The value of IBE is expressed as a percentage, and the range of the value is ±10%.

### 2.3.2. Entropy Water Quality Index

The water quality index (WQI) and entropy water quality index (EWQI) are significant methods to evaluate the suitability of groundwater for drinking purposes [31,32]. On the whole, the EWQI gives the more accurate and considered entropy of each parameter to calculate the index value for each sample location. In the EWQI, entropy information and entropy weight are important parameters reflecting the influence of various sources of contamination on the overall quality of water for drinking purposes; also, it reduces a larger set of data into the comprehensive and informative value for each sample location [6]. The five-step approach was adopted to estimate the EWQI of groundwater in the study area, as shown in Figure 2.

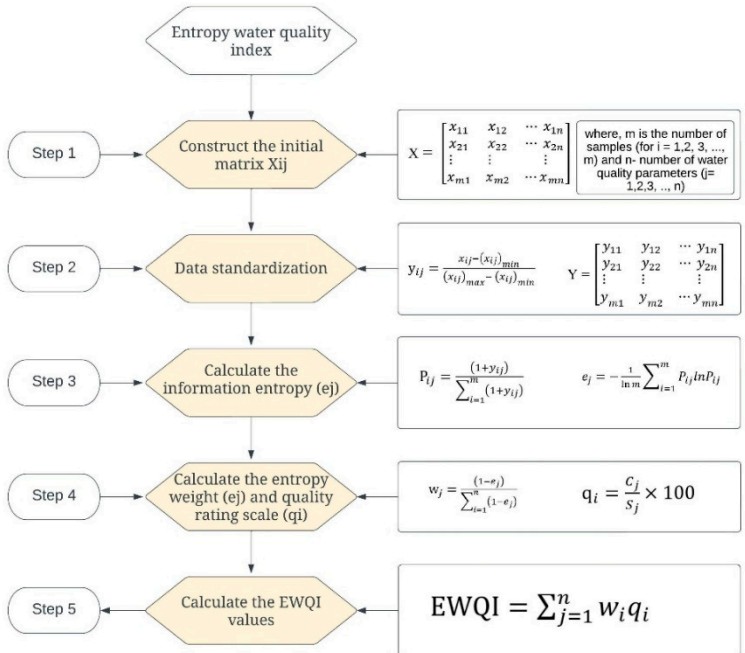

**Figure 2.** Framework to calculate entropy water quality index.

### 2.3.3. Nitrate Pollution Index

The NPI is a significant method to assess the contamination level of excess nitrate in groundwater and estimate the impact on human health [49,50]. El Mountassir et al. [51] developed the NPI method to classify groundwater quality, where a value less than 0 is clean, and 0–1, 1–2, 2–3 and greater than 3 are light pollution, moderate pollution, significant pollution and very significant pollution, respectively. The NPI is estimated using Equation (2):

$$\text{NPI} = \frac{C_S - \text{HAV}}{\text{HAV}} \tag{2}$$

where $C_s$ is the concentration of nitrate in groundwater samples, and HAV is the threshold concentration of nitrate in groundwater due to anthropogenic activities (20 mg/L).

### 2.3.4. Human Health Risk Evaluation (HHRE)

The rapid increase in population, urbanization and industrialization are major threats to human health. They increase the groundwater demand daily [21,37,52–54]. The continuous consumption of groundwater has major impacts on human health, such as carcinogenic and non-carcinogenic risks. The HHRE is an effective way to evaluate the potential impact on human health of consuming contaminated groundwater. A value of the hazards quotient (HQ) greater than one indicates a risk, and less than one is safe for drinking purposes. Figure 3 elaborates the steps to be followed to calculate the HQ value for each groundwater sample in the study region.

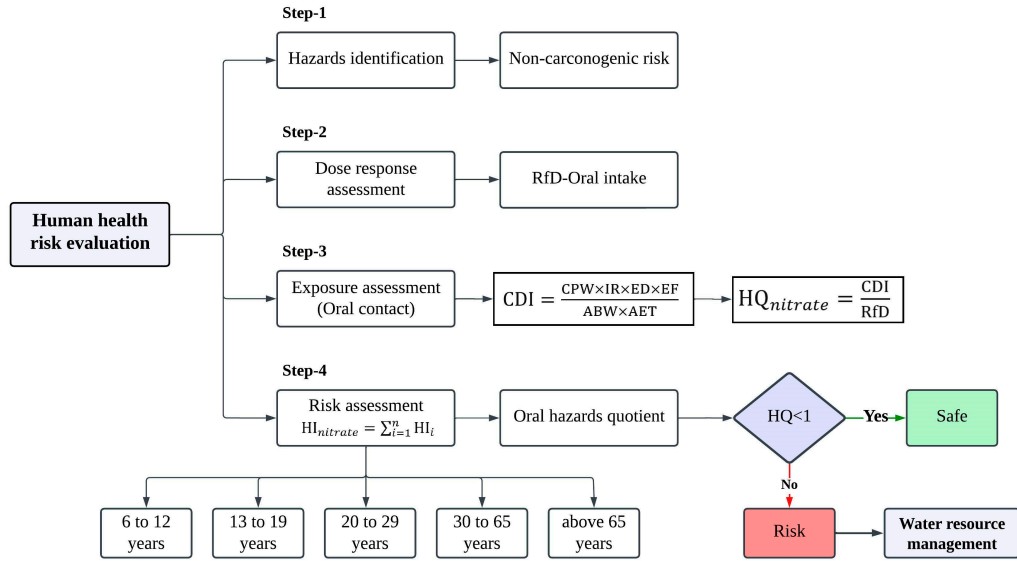

**Figure 3.** Step-by-step procedure for calculating human health risk.

### 2.3.5. Irrigation Indices

In the study region, agriculture is the major source of income for the people; consequently, the quality of groundwater for irrigation purposes also needs to be evaluated before use [1,55–59]. Contaminated groundwater used for agricultural purpose causes a very low crop yield, crop damage, crop diseases, crop growth problems, etc. Plus, it indirectly affects the health of humans who consume food from the contaminated zone. Various indices, such as the sodium absorption ratio (SAR), percentage sodium (%Na), residual sodium carbonate (RSC), magnesium absorption ratio (MAR), permeability index (PI) and Kelly ratio (KI), were calculated to assess the quality of groundwater for irrigation purposes. The formulas used to compute those indices in this study are shown in Figure 4.

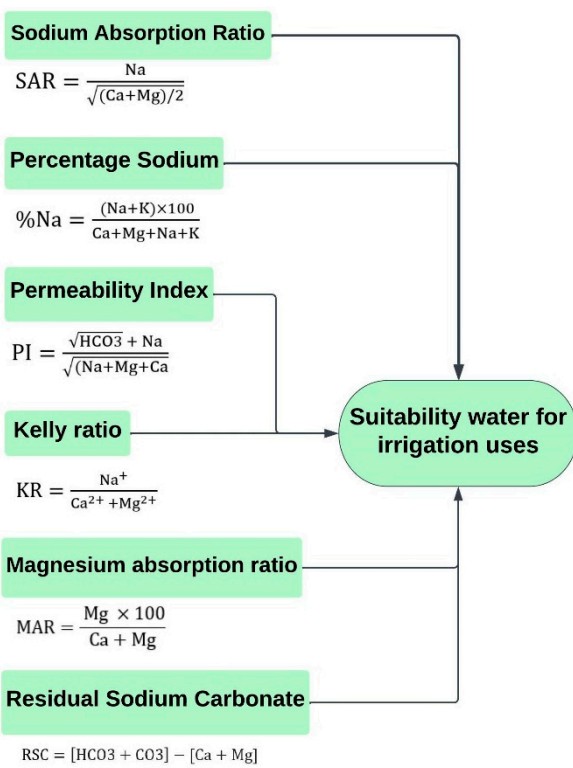

**Figure 4.** Irrigation indices methodology.

### 2.3.6. Principal Component Analysis (PCA)

PCA is a significant and widely used technique to analyze a large dataset containing a high number of dimensions per observation and enhance the interpretability of data. In general, PCA is the best statistical technique to reduce the dimensionality of a dataset and convert it into the new coordinate system with fewer dimensions than the initial dataset [56]. PCA with K-means clustering is the best approach to differentiate natural from anthropogenic sources of groundwater contamination and conduct accurate data classification. In the present study, detailed PCA was carried out to identify the interrelationship between the water quality parameters using IBM SPSS software, version 20 (IBM Corp.).

### 2.3.7. Identification of Contaminant Sources

Geogenic and non-geogenic activities are the primary sources of dissolved ions in groundwater chemistry. Precipitation, dissolution, ion exchange and reverse ion exchange are geogenic sources that majorly influence the groundwater equilibrium. Meanwhile, non-geogenic sources such as residential waste disposal, dumping yards, underground pipeline leakages, fertilizers, pesticides and improper management of waste disposal also dissolve ions in groundwater. In the present study, both geogenic and anthropogenic activities affected the chemical equilibrium of groundwater, meaning it was essential to evaluate the sources of dissolved ions in the aquifer system using chloro-alkaline indices (CAI-I and CAI-II), a saturation index and bivariate diagrams. The chloro-alkaline indices are widely used to evaluate the role of the ion exchange process and cation exchange in an aquifer system. In this study, the CAI-I and CAI-II were calculated using Equations (3) and (4) proposed by Schoeller [60]. A positive value of the CAI index indicates a reverse ion exchange process, such as the release of $Ca^{2+}$ and $Mg^{2+}$ from the groundwater and absorption of $Na^+$ and $K^+$ on the aquifer stratum. In contrast, a negative value of the CAI index represents a base ion exchange process, such as the exchange of $Ca^{2+}$ and $Mg^{2+}$ ions from the groundwater with $Na^+$ and $K^+$ ions from the aquifer stratum.

$$CAI - I = \frac{Cl^- - (Na^+ + K^+)}{Cl^-} \tag{3}$$

$$CAI - II = \frac{Cl^- - (Na^+ + K^+)}{SO_4^{2-} + HCO_3^- + CO_3^{2-} + NO_3^-} \tag{4}$$

The saturation index is a significant index value to predict the chemical reaction of a particular mineral in groundwater chemistry using Equation (5). If the value of SI is zero, this indicates a stable equilibrium condition between water and specific minerals; if SI is greater than zero, this represents oversaturation of chemical activities of a specific mineral; and if SI is below zero, this indicates undersaturation of particular minerals.

$$SI = \frac{K_{IAP}}{K_{SP}} \tag{5}$$

The ionic bivariate diagram effectively represents the sources of dissolved ions in groundwater. In the present study, the geogenic source of contamination was identified by a plotting bivariate diagram of $Ca^{2+}+Mg^{2+}$ vs. $Na^++K^+$, $Ca^{2+}+Mg^{2+}$ vs. total cations, $Na^+$ vs. $Cl^-$, $Ca^{2+}+Mg^{2+}$ vs. $HCO_3^-$, $Ca^{2+}+Mg^{2+}$ vs. $HCO_3^-+SO_4^{2-}$ and $HCO_3^-$ vs. $Cl^-+SO_4^{2-}$. Meanwhile, the non-geogenic source was identified by a bivariate diagram of $NO_3^-+Cl^-/HCO_3^-$ vs. TDS. Contaminants were identified to support remedial measures in the specific location, to enhance the sustainable practices in the rural environment.

## 3. Results and Discussion

### 3.1. Hydrochemical Composition of Groundwater

In the present study, hydrochemical characteristics of the groundwater results (Table 1) showed that salinity and weathering of parent rocks influence the groundwater quality in a few sample locations [61,62]. Excess amounts of $Ca^{2+}$, $Mg^{2+}$, $Na^+$ and $K^+$ indicate that ion exchange processes and evaporation play major roles in a few sample locations [63–65]. The excess chloride and sulfate in groundwater were due to weathering of parent rocks and soil, the nature of aquifers, the dissolution of aquifer minerals and wastewater disposed into open land [66–68]. The weathering of dolomite and calcite rocks in the study area is a major source of excess bicarbonate in groundwater. This shows that anthropogenic activities such as the use of uncovered septic tanks and synthetic fertilizers are important sources of nitrate in groundwater [69–73].

**Table 1.** Statistical analysis of groundwater samples in study region.

|  | Minimum | Maximum | Mean | Kurtosis | Skewness | WHO 2011 | % of the Sample Exceeds |
|---|---|---|---|---|---|---|---|
| pH | 7.30 | 8.54 | 7.77 | −0.27 | 0.49 | 6.5–8.5 | 0.00 |
| TDS | 100.30 | 985.00 | 511.60 | −1.43 | −0.03 | 1000 | 0.00 |
| TH | 50.00 | 640.31 | 268.76 | −0.79 | 0.68 | 500 | 13.95 |
| EC | 96.00 | 2023.00 | 931.38 | −0.37 | 0.32 | 1500 | 9.30 |
| $Ca^{2+}$ | 60.00 | 250.00 | 126.99 | 2.04 | 0.74 | 200 | 4.65 |
| $Mg^{2+}$ | 12.00 | 174.00 | 84.23 | −0.29 | 0.10 | 150 | 6.97 |
| $Na^+$ | 13.00 | 457.00 | 157.12 | 1.56 | 0.99 | 200 | 18.60 |
| $K^+$ | 0.00 | 64.46 | 34.29 | −1.35 | 0.53 | 12 | 27.90 |
| $Cl_2^-$ | 12.50 | 716.00 | 239.62 | 2.91 | 1.44 | 250 | 32.55 |
| $HCO_3^-$ | 82.00 | 769.00 | 237.86 | 4.37 | 1.90 | 500 | 4.65 |
| $SO_4^-$ | 7.00 | 421.00 | 154.93 | 1.07 | 0.60 | 250 | 9.30 |
| $NO_3^-$ | 21.00 | 64.00 | 40.53 | −1.31 | 0.24 | 50 | 37.20 |
| $F^-$ | 0.22 | 1.18 | 0.67 | −1.25 | 0.07 | 1.5 | 0.00 |

### 3.2. EWQI

The values of the EWQI of the groundwater samples in the study area ranged between 26.35 to 198.56, with a mean of 103.48. In the EWQI classification, 4 samples (9.30%) were excellent, 20 samples (46.51%) were good, 12 samples (27.91%) were medium and 7 samples (16.28%) were poor quality for drinking purposes (Table 2). Based on the spatial analysis, contaminated zones were categorized and identified in the study region. To delineate the

area by contaminants, 3.96 sq.km has excellent-, 285.92 sq.km has good-, 304.98 sq.km has medium- and 16.84 sq.km has poor-quality water (Figure 5). The results specify that a total of 321.82 sq.km of the area needs to be monitored before use.

**Table 2.** Entropy water quality index in the study area.

| EWQI | Class of Water | Sample Count | % of Samples | Area Occupied |
|------|----------------|--------------|--------------|---------------|
| <50 | Excellent | 4 | 9.30 | 3.96 |
| 50–100 | Good | 20 | 46.51 | 285.92 |
| 100–150 | Medium | 12 | 27.91 | 304.98 |
| 150–200 | Poor | 7 | 16.28 | 16.84 |
| >200 | Extreme poor | 0 | 0.00 | 0.00 |

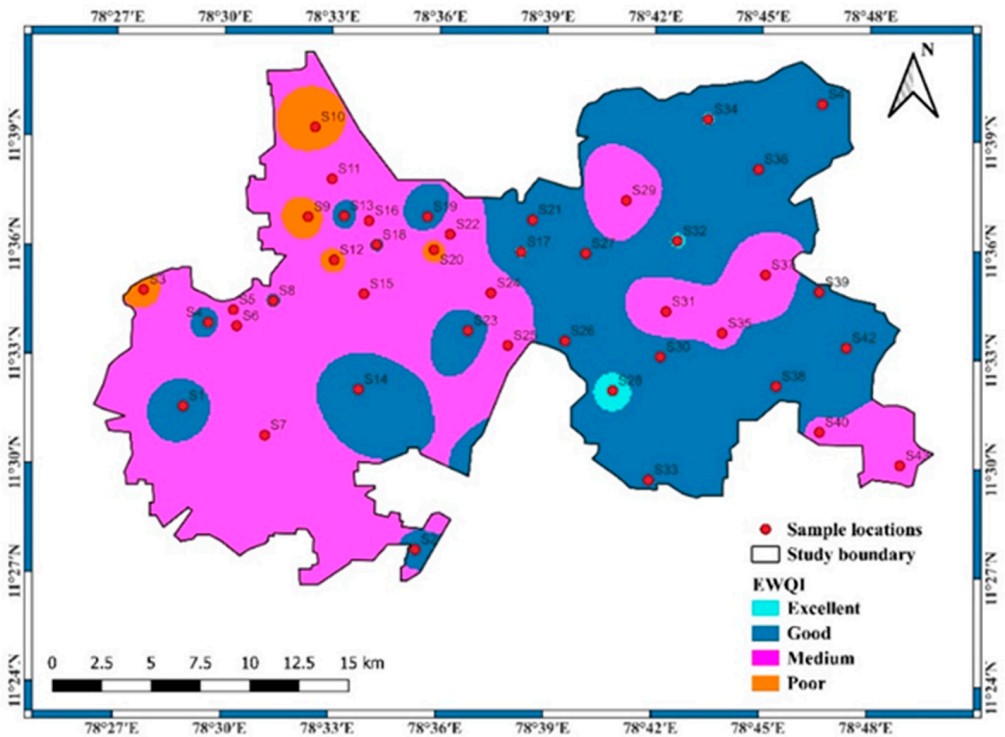

**Figure 5.** Irrigation indices methodology.

*3.3. NPI*

The spatial analysis of nitrate concentrations showed that 564.04 sq.km of area is suitable and 47.67 sq.km is unsuitable for drinking purposes. The northwest and southeast were identified as highly contaminated zones due to their nitrate concentrations (Figure 6). During the sample collection and preliminary survey, we found that agricultural activities were extensive in the southeast zone, and waste disposal was found in the study region's northwest zone. The NPI value of groundwater samples shows that 58.1% of samples had light pollution, while 27.9% and 13.9% had moderate or significant pollution, respectively (Table 3). The results showed that anthropogenic activities are the primary source of excess nitrate contamination in the study region.

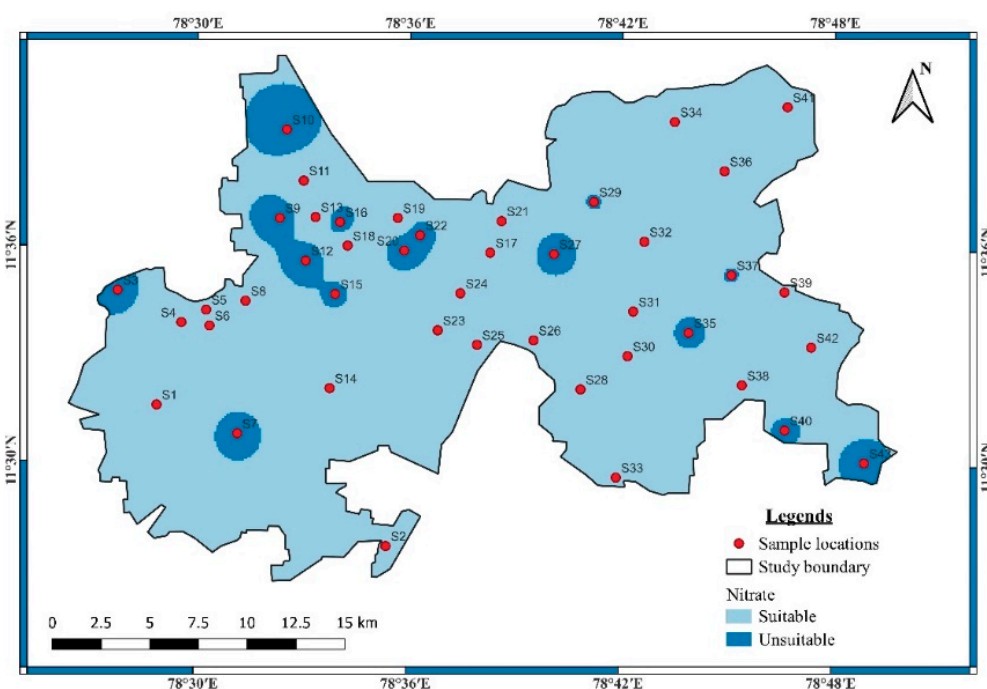

**Figure 6.** Nitrate contamination zones in the study area.

**Table 3.** NPI classification of groundwater samples.

| NPI Value | Contamination Type | No. of Samples | % of Samples |
|---|---|---|---|
| <0 | Clean | 0 | 0.00 |
| 0 to 1 | Light pollution | 25 | 58.14 |
| 1 to 2 | Moderate pollution | 12 | 27.91 |
| 2 to 3 | Significant pollution | 6 | 13.95 |
| >3 | Very Significant pollution | 0 | 0.00 |

### 3.4. HHRE

The nitrate contamination analysis and NPI results showed that 41.86% of the samples were contaminated due to excess nitrate concentrations in groundwater. Hence, it was important to evaluate the non-carcinogenic risk to human health in the study area. In the present study, we categorized the study area population into five groups: 6–12 years, 13–19 years, 20–29 years, 30–65 years and over 65 years (Figure 7).

#### 3.4.1. Effect on 6–12-Year-Olds

The hazard quotient for the group aged 6–12 years ranged from $5.94 \times 10^{-01}$ to $1.84 \times 10^{+00}$ with a mean of $1.15 \times 10^{+00}$. The spatial analysis of exposure to nitrate contamination showed that 463.50 sq.km of area is at risk and 148.22 sq.km of area is safe (Figure 7a). The results indicate that the presence of nitrate causes serious health issues, such as low immunity and low body weight. In the study region, in 75.76% of the sample locations, we found health issues in children aged 6–12 years.

#### 3.4.2. Effect on 13–19-Year-Olds

The hazard quotient for the group aged 13–19 years ranged from $4.43 \times 10^{-01}$ to $1.35 \times 10^{+00}$ with a mean of $8.55 \times 10^{-01}$. The spatial analysis of exposure to nitrate contamination showed that 82.49 sq.km of area is at risk and 529.23 sq.km of area is safe. The research found that the northeast, southwest and a few locations in the central part of the study region are contaminated due to high concentrations of nitrate in groundwater (Figure 7b). In total, in 13.48% of the sample locations, we found health issues for teenagers.

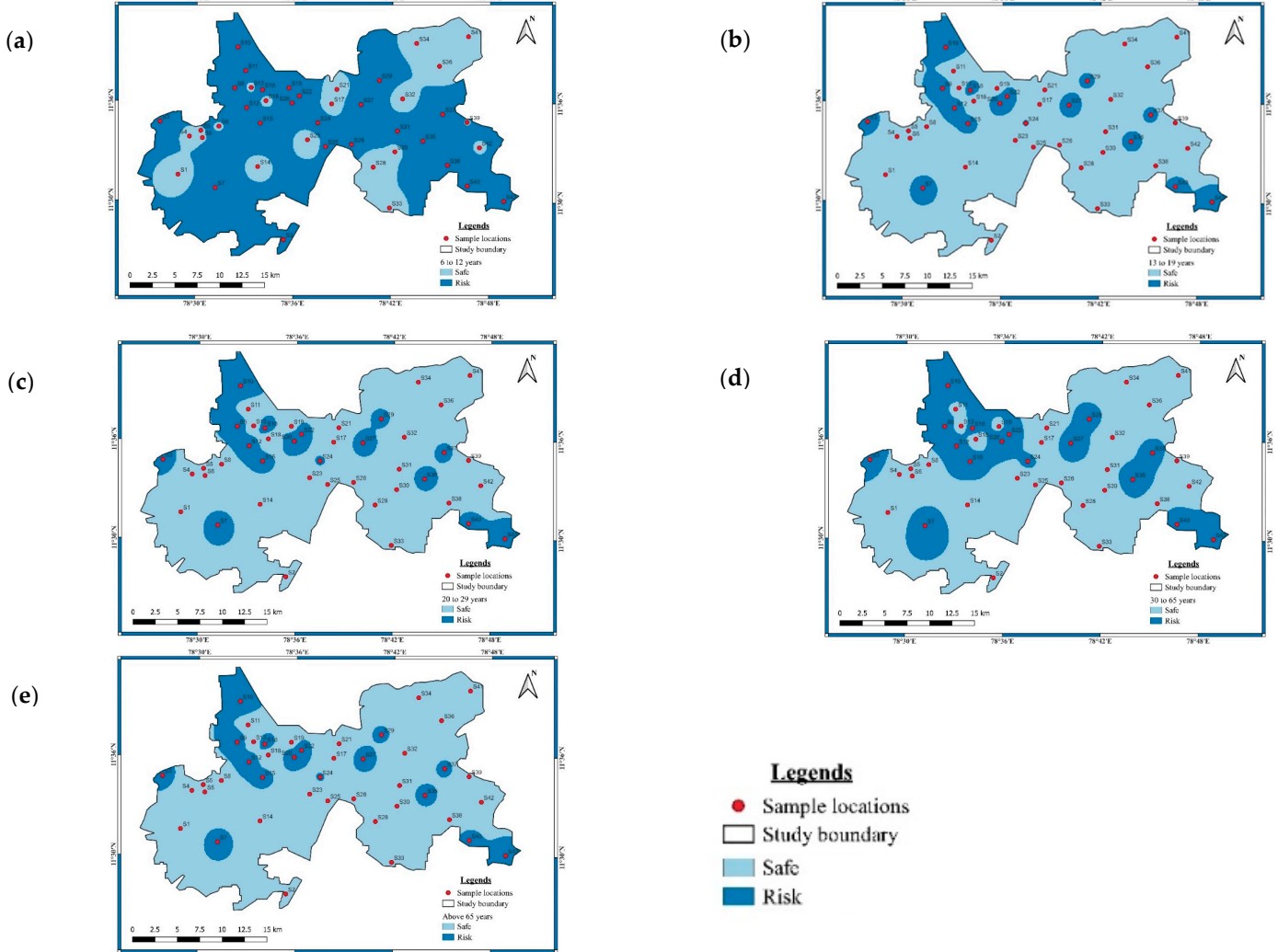

**Figure 7.** Non-carcinogenic risks for those aged: (**a**) 6–12 years; (**b**) 13–19 years; (**c**) 20–29 years; (**d**) 30–65 years; (**e**) over 65 years.

### 3.4.3. Effect on 20–29-Year-Olds

The hazard quotient for the group aged 20–29 years varied from $4.57 \times 10^{-01}$ to $1.39 \times 10^{+00}$ with a mean of $8.82 \times 10^{-01}$. The spatial analysis of exposure to nitrate contamination (Figure 7c) showed that 105.37 sq.km of area is at risk and 506.35 sq.km of area is safe. About 7 sample locations in the northeast, 4 sample locations in the southwest and 5 sample locations in the central part of the study area are contaminated due to excess nitrate. In the study region, in 17.22% of the sample locations, we found health issues affecting people aged 20–29 years.

### 3.4.4. Effect on 30–65-Year-Olds

The hazard quotient for the group aged 30–65 years ranged from $4.92 \times 10^{-01}$ to $1.50 \times 10^{+00}$ with a mean of $9.50 \times 10^{-01}$. The spatial analysis of exposure to nitrate contamination (Figure 7d) showed that 175.29 sq.km of area is at risk and 436.43 sq.km of area is safe. Most samples from the northeast, two from the southwest and four from the central part were contaminated. In the study region, in 28.65% of the sample locations, we found health issues affecting people aged 30–65 years.

### 3.4.5. Effect on People Aged over 65 Years

The hazard quotient for the group aged over 65 years ranged from $4.50 \times 10^{-01}$ to $1.37 \times 10^{+00}$ with a mean of $8.69 \times 10^{-01}$. The spatial analysis of exposure to nitrate contamination (Figure 7e) showed that 93.60 sq.km of area is at risk and 518.12 sq.km of area is safe. Contaminated zones were identified in the northeast and southwest parts of the study region. In the study region, in 15.30% of the sample locations, we found health issues affecting people aged over 65 years.

The results indicate that 37.5% of the sample locations pose a greater risk for children aged 6–12 years compared to older age groups (teenage, adult and elderly) in the study area (Figure 8). During the sample collection and preliminary investigation, the research team recorded the primary sources of contamination in the study area. Municipal waste disposal, improper maintenance of the sewerage system and leachates from open dumping yards were found in the central part of the study area. The usage of synthetic fertilizers and pesticides and agricultural waste disposal were found in the northeast and southwest parts of the study region. The key natural source of nitrate contamination was the decomposition of organic contaminants in the soil, and it was noted that water–rock interaction may also have released minerals that threw the groundwater chemistry out of equilibrium.

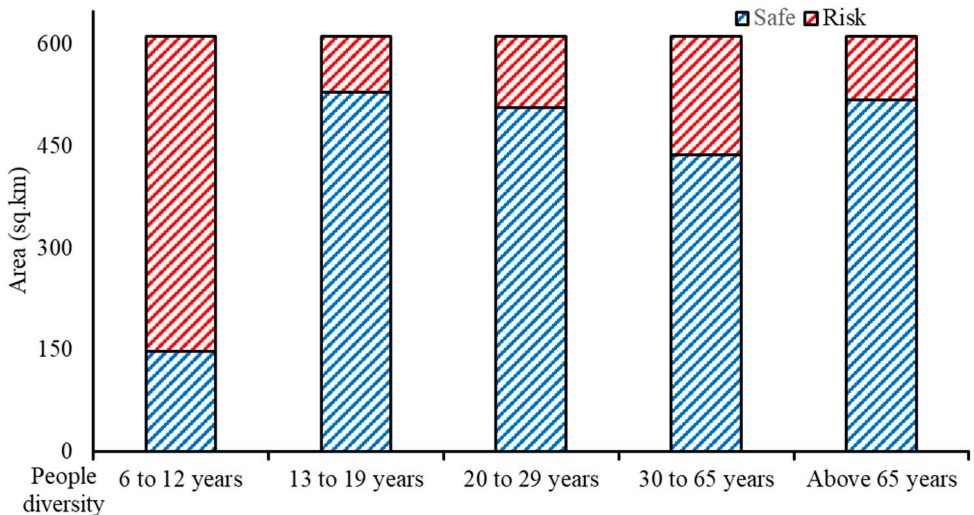

**Figure 8.** Non-carcinogenic risks for various groups of people in the study area.

### 3.5. Irrigation Indices

The analysis revealed that all the samples were excellent for irrigation. Based on their SAR classification, 11.63%, 53.49% and 34.88% of the samples were excellent, good or permissible for irrigation based on the percentage of sodium, respectively. Meanwhile, 97.67% of samples were satisfactory and 2.33% were marginally polluted based on the residual sodium carbonate. The magnesium absorption ratio revealed that 41.86% of the samples were suitable and 57.14% of the samples were unsuitable for irrigation, while 90.7% of the samples were class II and 9.3% of the samples were class III based on the permeability index. Overall, 88.37% of the samples were good and 11.63% of the samples were doubtful for irrigation use in the study area (Table 4). The irrigation indices revealed that magnesium and sodium are excessive in the study area, meaning the groundwater needs to be monitored before it is used. The processes of ion exchange and reverse ion exchange, weathering of parent rock, mineral dissolution of aquifers, evaporation and leachate from the municipal waste dumping yards and infiltration of rainfall are the primary sources of groundwater contamination in the study area.

**Table 4.** Classification of groundwater based on irrigation indices.

| Range | No. of Samples | Samples (%) | Class |
|---|---|---|---|
| | Sodium absorption ratio | | |
| Less than 10 | 43 | 100 | Excellent |
| 10–18 | 0 | 0 | Good |
| 18–26 | 0 | 0 | Doubtful |
| Greater than 26 | 0 | 0 | Unsuitable |
| | Percentage of sodium | | |
| 0–20 | 5 | 11.63 | Excellent |
| 20–40 | 23 | 53.49 | Good |
| 40–60 | 15 | 34.88 | Permissible |
| 60–80 | 0 | 0.00 | Doubtful |
| Greater than 80 | 0 | 0.00 | Unsuitable |
| | Residual sodium carbonate | | |
| Less than 1.25 | 42 | 97.67 | Satisfactory |
| 1.25–2.5 | 1 | 2.33 | Marginal |
| Greater than 2.5 | 0 | 0.00 | Unsatisfactory |
| | Magnesium absorption ratio | | |
| Less than 50 | 18 | 41.86 | Suitable |
| Greater than 50 | 25 | 58.14 | Unsuitable |
| | Permeability index | | |
| Greater than 75 | 0 | 0.00 | Class I |
| 75–25 | 39 | 90.70 | Class II |
| Less than 25 | 4 | 9.30 | Class III |
| | Kelly ratio | | |
| Less than 1 | 38 | 88.37 | Good |
| 1–2 | 5 | 11.63 | Doubtful |
| Greater than 2 | 0 | 0.00 | Unsuitable |

*3.6. Statistical Analysis*

PCA of groundwater showed that six principal components could describe a total variance of 72.49%: 20.33% in PC1, 12.43% in PC2, 11.13% in PC3, 10.23% in PC4, 9.68% in PC5 and 8.69% in PC6. We categorized principal components greater than 0.75, 0.75–0.5 and 0.5–0.3 as strong, moderate and weak loading factors [48]. In factor 1, $Ca^{2+}$, $Mg^{2+}$, $Na^+$, $Cl^-$ and $SO_4^{2-}$ were strongly or moderately loaded, indicating that natural weathering process, ion exchange process, rock–water interaction and chemical dissolution influenced groundwater quality (Table 5). In factor 2, $Cl^-$ and $SO_4^{2-}$ were negatively loaded, reflecting the rich anthropogenic activities in the study area. Factors 3–6 suggested that alkali and alkaline earth metal weathering, nonpoint sources of pollution from agriculture fields, dissolution of carbonate minerals and bacterial decomposition of organic substances present in the soil were the major sources of contamination in the study area.

**Table 5.** Principal component analysis of groundwater samples.

| Variables | Principal Component (PC) | | | | | |
|---|---|---|---|---|---|---|
| | 1 | 2 | 3 | 4 | 5 | 6 |
| pH | 0.06 | −0.23 | 0.49 | 0.40 | 0.52 | −0.05 |
| TDS | −0.24 | 0.56 | 0.18 | −0.22 | 0.16 | 0.28 |
| TH | 0.23 | 0.34 | 0.48 | 0.12 | 0.03 | −0.05 |
| EC | 0.14 | −0.35 | −0.39 | 0.42 | −0.10 | 0.45 |
| $Ca^{2+}$ | 0.59 | 0.27 | −0.19 | 0.30 | −0.15 | −0.35 |
| $Mg^{2+}$ | 0.55 | 0.50 | −0.19 | 0.25 | −0.18 | 0.06 |
| $Na^+$ | 0.71 | −0.11 | −0.01 | −0.53 | 0.22 | 0.28 |

**Table 5.** *Cont.*

| Variables | Principal Component (PC) | | | | | |
|---|---|---|---|---|---|---|
| | 1 | 2 | 3 | 4 | 5 | 6 |
| $K^+$ | 0.28 | 0.33 | −0.09 | 0.48 | 0.38 | −0.10 |
| $Cl^-$ | 0.86 | −0.07 | −0.20 | −0.28 | −0.10 | −0.02 |
| $HCO_3^-$ | 0.04 | 0.03 | −0.43 | −0.03 | 0.75 | 0.25 |
| $SO_4^{2-}$ | 0.62 | −0.15 | 0.63 | −0.10 | 0.02 | 0.08 |
| $NO_3^-$ | 0.29 | −0.57 | 0.14 | 0.34 | −0.17 | 0.25 |
| $F^-$ | 0.14 | −0.418 | −0.19 | −0.18 | 0.27 | −0.68 |
| Total | 2.64 | 1.62 | 1.45 | 1.33 | 1.26 | 1.13 |
| % variance | 20.33 | 12.43 | 11.13 | 10.23 | 9.68 | 8.69 |
| Cumulative % | 20.33 | 32.76 | 43.89 | 54.12 | 63.80 | 72.49 |

*3.7. Geogenic Sources*

3.7.1. Rock Weathering's Dominance of Groundwater Chemistry

Rock–water interaction, parent rock weathering (cation exchange) and halite dissolution play a vital role in dissolving ions into aquifer systems in the study area. The process of halite ions' dissolution releases $Na^+$ and $Cl^-$ ions in excess concentrations, driving the processes of rock–water interaction and weathering of rocks. In the present study, 48.83% of the samples were found to be below the theoretical line for $Na^+$ and $Cl^-$ ions specifying the reverse ion exchange process. Meanwhile, just over half of the sample locations (51.17%) were above the theoretical line, indicating that halite dissolution, rock–water interaction and weathering of parent rock are the leading sources of contamination (Figure 9a). The study was conducted in a region with a tropical climate (semi-arid) and where agriculture is the primary source of income for people. The results of this study indicate that waste from agriculture fields and irrigated runoff play a significant role in supporting the excess concentration of $Cl^-$ ions. The following reactions (Equations (6) and (7)) are the processes through which excess $Na^+$ and $Cl^-$ concentrations appear in groundwater:

$$\text{NaCl halite dissolution} \rightarrow Na^+ + Cl^- \tag{6}$$

$$(Ca^{2+}, Mg^{2+}, K^+, Na^+) \text{ silicates} + \text{rock weathering } (H_2CO_3) \rightarrow Ca(HCO_3)_2 + Mg(HCO_3)_2 \\ + NaHCO_3 + KHCO_3 + Ca(HCO_3)_2 + Mg(HCO_3)_2 + H_4SiO_4 + \text{clay product} \tag{7}$$

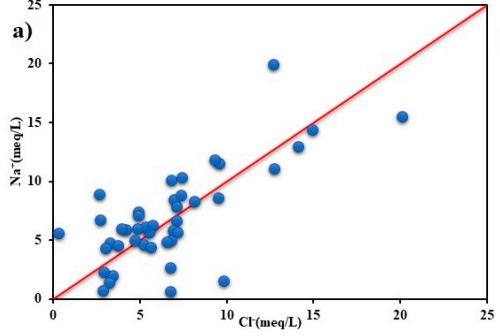
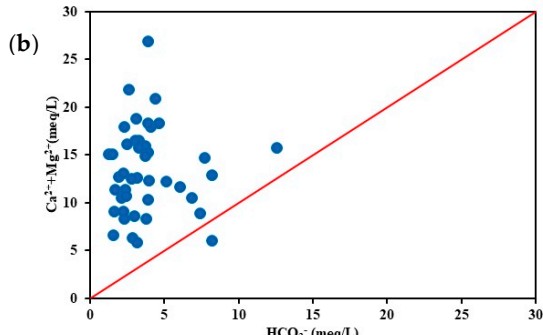

**Figure 9.** Ionic relationships of (**a**) $Na^+$ vs. $Cl^-$ and (**b**) $(Ca^{2+}+Mg^{2+})$ vs. $HCO_3^-$.

The ionic bivariate diagram of $Ca^{2+}+Mg^{2+}$ vs. $HCO_3^-$ helps to represent how rock weathering (cation exchange) highly influences the groundwater chemistry. Figure 9b shows that 98% of the sample locations were above the 1:1 equiline of $Ca^{2+}+Mg^{2+}$ vs. $HCO_3^-$; this was caused by rock–water interaction and rock weathering, which are the dominant processes in the study area. The finding indicates that major minerals such as $Ca^{2+}+Mg^{2+}$ and $HCO_3^-$ resulted from carbonate (calcite) rocks, as proven by plotting the bivariate diagram of $HCO_3^-$ vs. $Cl^-+SO_4^{2-}$ (Figure 10a). About 88.37% of the sampling points were found to be above the 1:1 line of $Ca^{2+}+Mg^{2+}$ vs. $HCO_3^-+SO_4^{2-}$ (Figure 10b),

which confirms the release of carbonate ions from weathering processes, along with con-taminated soil, which equally supplies ions to the groundwater system. Excess $Ca^{2+}+Mg^{2+}$ confirmed the process of reverse ion exchange owing to the origin and nature of the aquifer stratum. In addition, the reverse ion exchange process was confirmed by plotting a $Ca^{2+}+Mg^{2+}$ vs. $Na^++K^+$ bivariate diagram. It was found that 13.95% of samples moved toward Na++K+ ions, and it was confirmed that $Ca^{2+}+Mg^{2+}$ made a major contribution to the groundwater chemistry (Figure 11a). All sample points (100%) in the study region were found below the 1:1 line of total cations vs. $Ca^{2+}+Mg^{2+}$ (Figure 11b), which affirmed that carbonate weathering is the major source of cation ions in the study region.

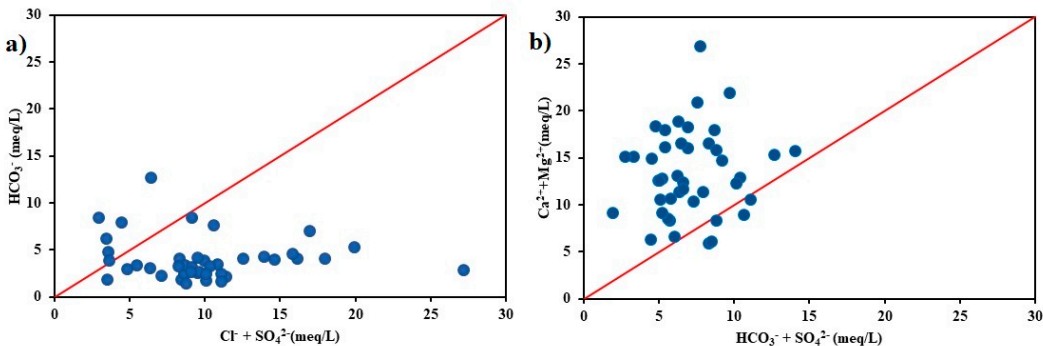

**Figure 10.** Ionic relationships of (**a**) $HCO_3^-$ vs. $(Cl^-+SO_4^{2-})$ and (**b**) $(Ca^{2+}+Mg^{2+})$ vs. $(HCO_3^-+SO_4^{2-})$.

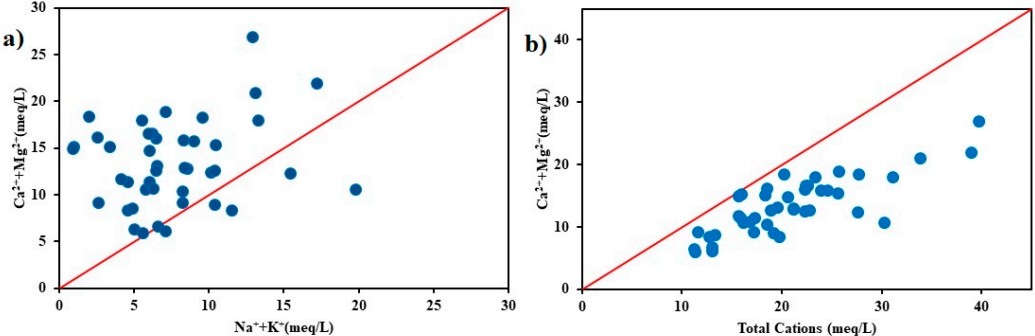

**Figure 11.** Ionic relationships of (**a**) $(Ca^{2+}+Mg^{2+})$ vs. $(Na^++K^+)$ and (**b**) $(Ca^{2+}+Mg^{2+})$ vs. total cations.

### 3.7.2. Ion Exchange's Dominance of Groundwater Chemistry

About 48.83% of the sample points had excess $Cl^-$ ions, which represents the contri-bution of the reverse ion exchange process in the aquifer stratum. Estimating the chloro-alkaline indices (CAI-I and CAI-II) of groundwater samples helped to validate the reverse ion exchange phenomenon in the study area. The calculated value of CAI-I varied from $-1.29$ to 1.85 with a mean of $-0.04$, and CAI-II ranged between $-2.85$ and 0.84 with a mean of $-0.24$. Figure 12 demonstrates that 65.11% of the samples were found in the negative indices of CAI-I and CAI-II, representing the dominance of the cation exchange process (Equation (8)). Meanwhile, 34.88% of groundwater points fell in the positive indices of CAI-I and CAI-II, representing the contribution of the reverse ion exchange process (Equation (9)). This reveals that the cation exchange process dominated the groundwater chemistry compared to the ion exchange process in the study area.

$$Ca^{2+} + 2Na^+ \rightarrow 2Na^+ + X_2 \tag{8}$$

$$2Na^+ + CaX_2 \rightarrow Ca^{2+} + 2NaX \tag{9}$$

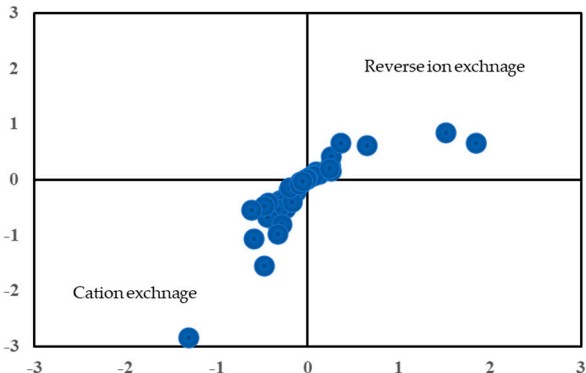

**Figure 12.** Ion exchange diagram of groundwater samples in the study region.

### 3.7.3. Evaporation's Governance of Groundwater Chemistry

The aquifer and soil in the study region were rich in carbonate ions, suggesting a predominantly tropical climatic (semi-arid) condition. This affects the possibility that evaporation and dissolution of minerals will take place in the groundwater system. It has significance for assessing the contribution of the evaporation process in groundwater chemistry. In the present study, the SI values of anhydrite, aragonite, dolomite, fluorite, calcite, halite and gypsum were computed and plotted, as shown in Figure 13. All groundwater sample locations were found to have positive values of SI concerning aragonite (2.31–3.70 with a mean of 2.91), calcite (2.46–3.84 with a mean of 3.05), dolomite (4.12–7.60 with a mean of 5.98) and fluorite (0.36–2.40 with an average of 1.43), indicating an oversaturated state (Table 6). Meanwhile, all sample points were found to have negative values of SI for halite (−4.80 to −2.58 with a mean of −3.56), indicating an undersaturated state (Figure 13).

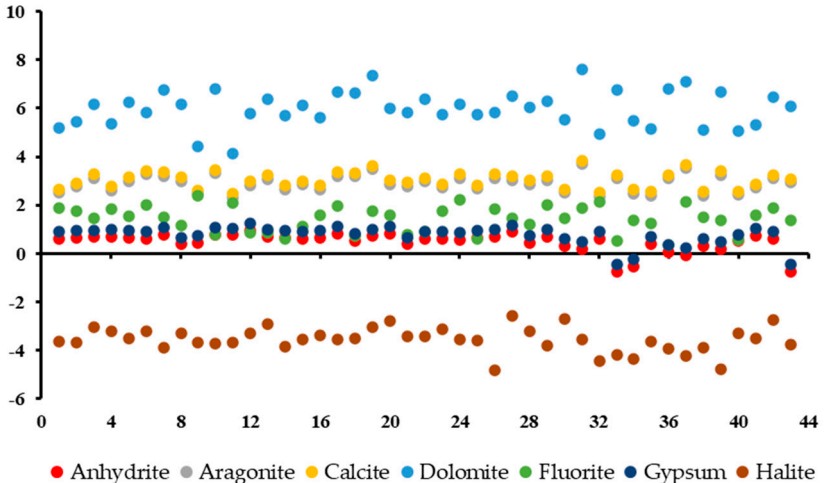

**Figure 13.** Ion exchange diagram of groundwater samples in study region.

### *3.8. Non-Geogenic Source*

As stated earlier, the study area comprises agricultural fields, and improper management of disposal of municipal waste, septic tanks and dumping yard leachates were the major issues identified in the study region. We sought to prove the anthropogenic sources of dissolved ions in the groundwater system by plotting a bivariate diagram of $NO_3^- + Cl^-/HCO_3^-$ vs. TDS. Figure 14 shows a linear trend of $y = -0.0008x + 1.865$, $R^2 = 0.062$, which clearly supports the dominance of anthropogenic activities in affecting the chemical composition of groundwater. Agricultural waste runoff, waste disposal in open land, municipal waste disposal and use of synthetic fertilizers and pesticides were the chief sources of anthropogenic contamination, as was noted during the sample collection and preliminary survey of the research area.

**Table 6.** Description statistical analysis of chloro-alkaline and saturation indices.

| Parameters | Minimum | Maximum | Mean |
|---|---|---|---|
| Chloro-alkaline index (CAI) | | | |
| CAI-I | −1.30 | 1.86 | −0.05 |
| CAI-II | −2.86 | 0.84 | −0.24 |
| Saturation index | | | |
| Anhydrite | −0.75 | 0.97 | 0.47 |
| Aragonite | 2.31 | 3.70 | 2.91 |
| Calcite | 2.46 | 3.84 | 3.05 |
| Dolomite | 4.12 | 7.60 | 5.98 |
| Fluorite | 0.36 | 2.40 | 1.43 |
| Gypsum | −0.46 | 1.26 | 0.76 |
| Halite | −4.80 | −2.58 | −3.56 |

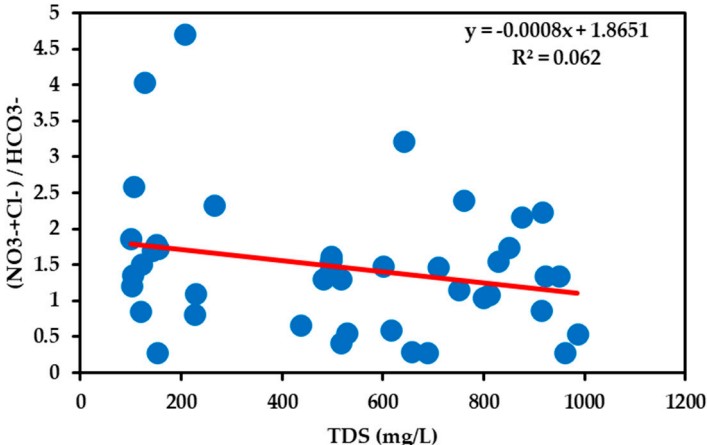

**Figure 14.** Ion exchange diagram of groundwater samples in study region.

## 4. Conclusions

The present study mainly aimed to assess the suitability of groundwater for drinking and irrigation purposes and to evaluate the non-carcinogenic risk to human health of nitrate contamination of groundwater. The research results showed that, based on the EWQI, 12 samples (27.91%) were medium- and 7 samples (16.28%) were poor-quality for drinking purposes, and they covered an area of 321.82 sq.km in the northwest and southeast zones of the study area. The NPI results further revealed that 27.91% of the samples had moderate pollution and 13.95% had significant pollution. Moreover, a non-carcinogenic risk assessment showed that 37.5% of the sample locations posed a risk for 6–12-year-old children more so than teenagers, adults and the elderly. Plus, the irrigation indices revealed that magnesium and sodium are in excess in the groundwater. The statistical relationships of groundwater samples showed that $Ca^{2+}$, $Mg^{2+}$, $Na^+$, $Cl^-$ and $SO_4^{2-}$ are strongly and moderately loaded. This indicates that the natural weathering process, ion exchange process, rock–water interaction and dissolution of chemicals influence the quality of groundwater. The results also showed that physical and chemical weathering of parent rocks, dissolution of aquifer minerals, the ion exchange process and rock–water interaction were the natural sources of groundwater contamination. Meanwhile, decomposition of organic substances, disposal of waste from the semi-urban area, improper waste management and dumping yards' leachates were the anthropogenic sources of contamination in the study area. The present research findings will help governmental authorities to take remedial measures to enhance sustainable practices to reduce groundwater contamination. The study also recommends introducing an artificial recharge site to improve the groundwater status in the study area. The future scope of the present research is to compare and integrate information on seasonal variations and the effects of the present anthropogenic activities

on groundwater quality, as well as add a detailed evaluation of the human health risk assessment in the present study.

**Author Contributions:** Conceptualization and formal analysis, B.P.; methodology and software, B.P., N.R., S.P.K. and S.K.; writing and editing, B.P., N.R. and B.B. All authors have read and agreed to the published version of the manuscript.

**Funding:** This research project was partially supported by Second Century Fund (C2F) no. 4/2564-Track A, Chulalongkorn University.

**Data Availability Statement:** The data are available on request from the corresponding author.

**Acknowledgments:** This publication was partially supported by Ratchadaphisek-somphot Endowment Fund no. FY-2564 and Second Century Fund (C2F) no. 4/2564- Track A, Chulalongkorn University.

**Conflicts of Interest:** The authors declare no conflict of interest.

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
