# Peer review of "Quality and Health Risk Assessment of Groundwater for Drinking and Irrigation Purpose in Semi-Arid Region of India Using Entropy Water Quality and Statistical Techniques"

_water, doi:10.3390/w15030601_

Round 1

Reviewer 1 Report

I have thoroughly gone through the paper entitled “Quality and health risk assessment of groundwater for drinking and irrigation purpose in semi-arid region of India using entropy water quality and statistical techniques”.

 I appreciate the authors for taking up this topic for research which is very much important for our country and other developing countries as well. In this paper scientific experiments have been performed properly following standard methods. 

My comments on the paper are provided below:

1.     Words used in the title should be avoided in the list of key words.

2.     Authors need to check the whole manuscript minutely for correcting minor mistakes.

3.     Lin no. 127 dense pollution? Not clear

4.     Line no. 191 to 197 repeated in line no. 205 to 211

5.      In line no.113 the word millet is written twice.

6.     Authors have performed PCA for proper interpretation of data.  I would appreciate  if authors perform Varimax rotation also. Because it is an important second step in PCA. It will further clarify the relationship among factors.

7.     For statistical analyses which package was used? It should be mentioned in the materials and methods.

8.     Moderate English language change is required.

Author Response

Response to Reviewer#1’s Comments

Notes: Our revised text regarding the reviewer#1’s comments are shown in dark red font in the revised manuscript.  Page/line numbers in reviewer’s comments refer to the original manuscript while line numbers in our responses refer to the revised manuscript.

  1. Words used in the title should be avoided in the list of key words.

Response: Thank you for your constructive comment. We replaced “entropy water quality index” which is used in the title with “human health” and “noncarinogenic risk” in the list of Key words as the reviewer suggested. (Line 34-35)

  1. Authors need to check the whole manuscript minutely for correcting minor mistakes.

Response: Thank you for your suggestion. The minor mistakes have been corrected in the revised manuscript.

  1. Lin no. 127 dense pollution? Not clear

Response: Thank you for pointing this out. It is a typo error. The word “dense pollution” has been corrected to “dense population.” (Line 133)

  1. Line no. 191 to 197 repeated in line no. 205 to 211

Response: Thank you for pointing this out. The repeat text in Line 205-211 has been removed.

  1. In line no.113 the word millet is written twice.

Response: Thank you for pointing this out. The repeat word “millet” has been removed from the manuscript. The new text reads:

 “The study region's major crops frequently cultivated are sugarcane, cotton, groundnut, gingelly, oilseeds, millet, and rice.” (Line 119-120)

  1. Authors have performed PCA for proper interpretation of data.  I would appreciate if authors perform Varimax rotation also. Because it is an important second step in PCA. It will further clarify the relationship among factors.

Response: Thank you for your constructive comment. We agree with the reviewer that Varimax rotation analysis is an important step in PCA, and we have analysed the relationship between each parameter using Varimax rotation in this study. Since our manuscript has contained 14 figures and 7 tables, we have not included the rotation matrix diagram in our manuscript due to number of figure and table restriction.

  1. For statistical analyses which package was used? It should be mentioned in the materials and methods.

Response: Thank you for your suggestion. We have added the information of the software used for statistical analyses in Section 2.3.6 in the revised manuscript. The additional text reads:

In the present study, detailed PCA has been carried to identify the interrelationship be-tween the water quality parameters using IBM SPSS software, version 20 (IBM Corp.).” (Line 204-206)

  1. Moderate English language change is required.

Response: English correction has been carried out in entire main text in the manuscript as the reviewer suggested.

Author Response

Response to Reviewer#2’s Comments

Notes: Our revised text regarding the reviewer#2’s comments are shown in green font in the revised manuscript.  Page/line numbers in reviewer’s comments refer to the original manuscript while line numbers in our responses refer to the revised manuscript.

  1. In the Abstract section, it is worth indicating the uniqueness/newness of the research conducted.

Response: Thank you for your constructive comment. The novelty and uniqueness of the present study has been added in Abstract. The additional text reads:

“The present study's novelty is integrating the hydrochemical analysis associated with entropy water quality index (EWQI), nitrate pollution index (NPI) and human health risk assessment.” (Line 22-24)

  1. The Introduction section does not provide an excellent theoretical introduction. The authors write about climatic differences in the Materials and Methods section (2.1). This thread is missing from the Introduction section. When writing about water quality, it is necessary to mention seasonality, which, according to many studies, affects the quality of ground and surface water. I suggest completing this thread in the Introduction section based on recent studies: https://doi.org/10.3390/en14133841 ; https://doi.org/10.1016/j.jhydrol.2020.125037 ; https://doi.org/10.1016/j.jhazmat.2020.124018

Response: Thank you for suggesting these articles. The details of seasonal variation and its effects on groundwater have been added in Introduction section. The additional text reads:

“Another factor, directly and indirectly, affecting groundwater quality is seasonal variability, including rainfall intensity, temperature variation, and humidity of atmospheres [40,41]. These are the primary factors that affect the chemical composition of groundwater during the process of evaporation during summer (dry season) and infiltration of rainfall water during rainy days (post-monsoon season) [42]. In the present research, the study area is a semi-arid region with a high experience of seasonal variability due to high temperature and evaporation process.” (Line 87-93).

  1. At the end of the introduction section, please provide the further structure of the manuscript in a few sentences. This will further improve its clarity.

Response: Thank you for this constructive comment. The additional text has been added to improve the clarity of the manuscript as followed:

“The novelty of the present objectives is the integrated approach to assessing the contaminants in groundwater. This is the first study carried out in the research region to assess groundwater quality for drinking and irrigation purposes. The structure of the present research is to carry out the EWQI of groundwater samples, nitrate contamination assessment aided with NPI, and human health risk assessment among various groups of people.” (Line 98-103).

  1. The second section lacks an indication of when the water samples were collected.

Response: Thank you for your suggestion. The period of data collection has been added in Section 2.3.1. The new text reads:

“The groundwater samples were collected in the dense population and agriculture field in the study area. A total of 43 samples were collected during the post-monsoon season before Covid-19 pandemic period (2018) in bore wells/hand pumps based on the available source of groundwater (Figure 1).” (Line 133-136)

  1. If the authors have such data, the effect of seasonality on water quality can be confirmed or negated in the discussion.

Response: The present study carried out the detailed analysis of hydrochemistry of groundwater during post monsoon period (one season). The comparison and effect of the seasonality variation in groundwater will be confirmed in the future research.

  1. At the end of the conclusions, please provide opportunities for future research in this area.

Response: Thank you for suggestion. The recommendation for further research works has been included in the Conclusion section as followed:

“The future scope of the present research is to compare and integrate the seasonal variation, effects of present anthropogenic activities on groundwater quality and detailed evaluation of human health risk assessment in the present study.” (Line- 484-486).
